# A Narrative Review of the Facts and Perspectives on Agricultural Fertilization in Europe, with a Focus on Italy

Arianna Latini [1,*], Germina Giagnacovo [1], Carlo Alberto Campiotti [1], Carlo Bibbiani [2] and Susanna Mariani [1]

1 Italian National Agency for New Technologies, Energy and Sustainable Economic Development (ENEA), Energy Efficiency Department, Casaccia Research Center, 00123 Rome, Italy; germina.giagnacovo@enea.it (G.G.); carloalberto.campiotti@enea.it (C.A.C.); susanna.mariani@enea.it (S.M.)
2 Department of Veterinary Sciences, University of Pisa, 56126 Pisa, Italy; carlo.bibbiani@unipi.it
* Correspondence: arianna.latini@enea.it

**Abstract:** Fertilizers stand at the base of current agricultural practices, providing the nutrient sustainment required for growing plants. Most fertilizers are synthetic chemicals, whose exploitation at very high levels poses a risk to cultivated land and the whole environment. They have several drawbacks including soil degradation, water pollution, and human food safety. Currently, the urgent need to counterbalance these negative environmental impacts has opened the way for the use of natural and renewable products that may help to restore soil structure, microorganism communities, nutrient elements, and, in some cases, to positively enhance carbon soil sequestration. Here, we endeavor to reinforce the vision that effective strategies designed to mitigate negative anthropic and climate change impacts should combine, in appropriate proportions, solutions addressed to a lower and less energy intensive production of chemicals and to a more inclusive exploitation of renewable natural products as biological soil amendments. After drawing an overview of the agricultural energy demand and consumption of fertilizers in Europe in the last few years (with a particular focus on Italy), this narrative review will deal with the current and prospective use of compost, biochar, and neem cake, which are suitable natural products with well-known potential and still-to-be-discovered features, to benefit sustainable agriculture and be adopted as circular economic solutions.

**Keywords:** fertilizers; biological soil amendment (BSA); organic farming; compost; biochar; neem cake

## 1. Introduction

Fertilizers are substances that provide soil with the nutrients required by plants to grow. Nowadays, mineral fertilizers worldwide represent the catalytic converter of intensive agriculture, allowing for high demand production to feed increasing populations. At the same time, it has now been ascertained that besides the main burdens for their industrial manufacture, such as considerable energy consumption [1] and high levels of pollutant generation and release to the environment [2], mineral fertilizers strongly and negatively impact the environment. More specifically, they lead to severe soil acidification [3], the impoverishment of rhizosphere biological activity in terms of microbiome richness and diversity [4,5], the presence of unsafe chemical residues in food products, etc. In order to face these occurrences and limit these negative consequences before it is too late, much research has addressed the application of natural, sustainable, and safe products that can, at least partially, replace mineral fertilizers.

Organic fertilizers refer to a wide range of natural products including manure and compost (other than slurry), peat, sewage sludge, humic acid, and several others. The use of organic fertilizers has recruited beneficial bacteria into the rhizosphere and reduced the available heavy metal content in soil and as in plant tissue [6]. Of course, the application of organic fertilizers must be planned with awareness, since an inappropriate distribution method and/or an overload could lead to environmental risks such as nitrate pollution [7].

In a different way, biofertilizers are based on beneficial microorganisms, including plant growth promoting rhizobacteria (PGPR) which can improve the nutrient (N, P, K, Fe, Zn) availability of soils thanks to their biological activity [8,9]. When applied in combination with the proper rates of mineral fertilizers, they can lead to maximum benefits in terms of yield and quality, allowing for a significant saving of synthetic chemicals [10]. Furthermore, an emerging promising alternative to mineral fertilizers in agriculture is represented by nanofertilizers. However, while nanofertilizers provide slow nutrient release and increase nutrient use efficiency, their extensive application to soil may pose further risks to the environment and to food safety [11].

In addition to these products, which exert the role of nutrient suppliers (indeed recognized as fertilizers) and have a direct effect on plant growth, there is a growing interest from farmers and researchers in the field on soil amendments, which can improve the physical status (i.e., the structure) of the soil and have an indirect effect on plant growth. A rapid regeneration of soil health, including a certain level of remediation from contaminants, may be achieved through the application of organic amendments obtained from (bio)transformation processes of agro-waste, food-waste, and food production by-products [12]. These products, also referred to as biological soil amendments (BSAs), present both fertilizing and amending properties. BSAs in soils have been shown to benefit microbially-mediated processes, even though the strong impact on ecosystem processes in long-term organic restoration may be different and requires an investigation in greater depth [13]. In particular, since manure and compost could represent a reservoir of pathogenic microorganisms, their application must be under strict control in order to limit pathogen prevalence in soils and, consequently, any possible fresh produce contamination [14].

The scenario for agricultural fertilization is very broad. Although the scientific literature offers numerous research articles and reviews, there is a shortage of studies that link together the different aspects of this topic, such as energy production costs and the different kinds of products, applications, and effects, in a comprehensive fashion. This manuscript, in the form of a narrative review, aims to examine the energy demand by the fertilizer industry, the consumption of fertilizers in the last several years, and the perspectives for the near future in the European Union and Italy as an EU member state case study. Afterwards, we introduce the current state of the art in the utilization of some BSAs that have grabbed a lot of attention, namely compost, biochar, and neem cake. We also refer to the current legislation that defines their application to agricultural soils. The content presented in this manuscript has high relevance for the development of strategies designed to mitigate negative anthropic and climate change impacts by combining, in appropriate proportions, solutions addressed to a less energy intensive production of chemicals and to a more inclusive exploitation of renewable natural products.

## 2. Methods

The structure of the current narrative review is based on the indications from Ferrari [15]. Data analyzed and reported proceed from the statistical office of the European Union (EUROSTAT, [16]) and The World Bank [17] and, for the specific case of Italy, from the Italian National Institute of Statistics (ISTAT, [18]). Information was extracted from publications and technical reports from the EU Science Hub [19]. The cited literature comes from NCBI PubMed [20], with a selection of studies considered more relevant for the context of this review, excluding those related to specific cultivated crops or specific geographic areas outside of the European context.

In order to establish the trends of fertilizer distribution, linear regression analysis was performed with Excel by creating an initial scatter plot and then adding the trendline.

## 3. Energy Requirement for the Industrial Production of Fertilizers

In 2018, according to EUROSTAT, the estimated final energy consumption by the agricultural sector in EU-27, mainly due to direct fuel and electricity costs, was 27.251 million tons

of oil equivalents (Mtoe, unit of energy describing energy content of all fuels when in a large scale), with the highest energy balances registered in France (4.089 Mtoe), Poland (3.918 Mtoe), Germany (3.342 Mtoe), the Netherlands (3.647), Italy (2.798 Mtoe), and Spain (2.458 Mtoe).

The Italian agriculture sector is currently among the best developed in Europe and is becoming more and more sustainable and energy efficient. Notwithstanding this, however, both direct and indirect energy consumption are still quite high (i.e., more than 5 Mtoe; [21]). In agriculture, in general, direct energy costs refer to the fuel consumed by tractors, agricultural machines, and by irrigation pumps. Indeed, the main potential and effective energy savings in this sector are related to irrigation. Moreover, and specifically concerning greenhouse production, direct energy is mainly required for climatization and humidity control [22]. In a different way, besides the synthesis and production of chemical pesticides, special supplements for feeding livestock, hybrid seeds, and other related products [23], the use of fertilizers represents the major indirect energy cost.

Energy consumption for producing, packing, and delivering the main types of mineral fertilizers (Table 1) can be substantial (e.g., up to 50 MJ per N kg for the urea in an average European plant) [1]. The main nitrogen components of fertilizers are the most energy-intensive to produce, while the P and K components all require less than 5 MJ/kg [24]. The manufacture of herbicides, fungicides, and insecticides entails even higher energy equivalents (Table 1). Most of the available energy data were available from older references. Thus, it is likely that current industrial plant consumption now requires a lower input. Surely, energy efficiency solutions, innovative and best available technologies, and continuous improvements of plant design have been, and will continue to be, fundamental and effective in order to consistently reduce the energy demand for fertilizer production.

Most NPK fertilizers, at a global level, are produced in some macro areas including East Asia, North America, Eastern Europe, and Central Asia [25]. It is worth noting that ammonia plants in Europe, on average, are considered the most energy efficient worldwide, immediately followed by those in the U.S.A., in contrast to lower energy efficiency in Russia and Ukraine (Fertilizers Europe, Brussels). To the fertilizer producers, low energy costs and therefore low environmental impact would be desirable. However, at the same time, both the market and agriculture need fertilizers with a long soil persistence that can confer optimal yields.

**Table 1.** Energy consumption, in megajoule per kilogram (MJ/kg), of the pure chemical element by the main types of mineral fertilizers.

| Type of Fertilizer | | | Primary Energy Consumption (MJ) | Reference |
|---|---|---|---|---|
| Grouping by Main Chemical | Name of Fertilizer | Common Abbreviation | | |
| N-based fertilizers (per kg of N) | Ammonia, $NH_4$ | A | 32 | [26] |
| | | | 36.6 | [27] |
| | | | 26.5–31.2 [(BAT) z] | [27] |
| | Ammonium nitrate, $NH_4NO_3$ | AN | 40 | [1] |
| | | | 29.8 | [1] |
| | | | $40.74 \pm 5.43$ | [28] |
| | Urea, $CO(NH_2)_2$ | Urea | 51.6 | [1] |
| | | | 44.1 [(BAT)] | [1] |
| | | | 22 | [26] |
| | | | 5.5 *–2.7 [(BAT) * y] | [29] |
| | Calcium ammonium nitrate | CAN | 42.6 | [1] |
| | | | 31.4 [(BAT)] | [1] |
| | Ammonium sulphate, $(NH_4)_2SO_4$ | AS | 42 | [1] |
| | | | ~6 | [26] |

**Table 1.** *Cont.*

| Type of Fertilizer | | | Primary Energy Consumption (MJ) | Reference |
|---|---|---|---|---|
| Grouping by Main Chemical | Name of Fertilizer | Common Abbreviation | | |
| P-based fertilizers (per kg of P) | Triple superphosphate, $Ca(H_2PO_4)_2$ | TSP | 30.5 | [1] |
| | | | 2.5–3 | [26] |
| | | | 15.15 | [30] |
| | Single superphosphate | SSP | 13 | [1] |
| | | | ~3 | [26] |
| | Phosphorus pentoxide, $P_2O_5$ | $P_2O_5$ | 12.44 | [31] |
| K-based fertilizers (per kg of K) | Muriate of potash, KCl | MOP | 10.6 | [1] |
| | | | ~3 | [26] |
| | Potassium oxide, $K_2O$ | PO | 11.15 | [31] |
| Limestone (per kg of Ca), $CaCO_3$ | | L | 2.3 | [1] |
| Herbicides (per kg) | | | 238 | [32] |
| | | | 298.9 (metolachlor, alacholor) | [33] |
| | | | 205.4 (atrazin) | [33] |
| Fungicides (per kg) | | | 216 | [32] |
| Insecticides (per kg) | | | 101.2 | [32] |

[z] (BAT) indicates production in a plant endowed with best available technology. Otherwise, energy consumption is referred to production in an average plant. [y] An asterisk (*) indicates urea per kg.

## 4. Consumption of Fertilizers in Traditional and in Organic Agriculture

Looking at the use of fertilizers in Italy as a measure of the amount of required nutrients for plants applied per unit of arable land, a significant decline was observed compared to twenty years ago (210.0 kg/ha in 1998 and 130.6 kg/ha in 2018). In the last decade it was always below the average of the European member states, which has settled at around 150 kg/ha in the last few years [34]. Based on the dataset reported in Table 2, Ireland represents the top EU fertilizer-using country, registering up to 1444.9 kg/ha in 2018, followed by Belgium, the Netherlands, and Slovenia, showing values close to 300 kg/ha. In the last twenty years, from 1998 to 2018, only a few EU countries (Denmark, Finland, Italy, the Netherlands, and Slovenia) showed a constant decline of fertilizer use every ten years. Some others (e.g., Belgium and Greece) showed a remarkable reduction in the middle, followed by an increase after ten years. Others (Ireland in particular, but also Bulgaria, Hungary, Latvia, Lithuania, Poland, Portugal, and Romania) showed an impressive increase which was related, at least partially, to the demand for higher agricultural production.

According to EUROSTAT, looking at the amounts of fertilizers spread over the EU countries by considering the sum of the total nitrogen and phosphorus, France, Germany, Spain, and Poland represent the EU member states spreading the largest fertilizer amounts to their land, with, respectively, 2.1, 1.3, 1.0, and 0.9 million tons of N, and 181.6, 87.8, 209.4, and 150.0 thousand tons of P, in 2019 [16]. The primacy of Ireland in the consumption of fertilizers is also evident when looking at the tons of nitrogen and the tons of phosphorus utilized for every 1000 inhabitants, respectively, 74 and 9 in 2019, showing a sharp decrease compared to twenty years earlier (Figure 1).

**Table 2.** Annual consumption of fertilizers [z] in kilogram per hectare (kg/ha) of agricultural land in the 27 member states of the European Union in 1998, 2008, and 2018.

| EU Member Country | Consumption of Fertilizers (kg/ha) | | | EU Member Country | Consumption of Fertilizers (kg/ha) | | |
|---|---|---|---|---|---|---|---|
| | 1998 | 2008 | 2018 | | 1998 | 2008 | 2018 |
| Austria | 175.9 | 110.0 | 135.1 | Italy | 210.0 | 143.5 | 130.6 |
| Belgium | 354.0 | 224.5 | 293.4 | Latvia | 46.5 | 66.9 | 101.2 |
| Bulgaria | 47.8 | 111.2 | 126.9 | Lithuania | 48.4 | 80.7 | 133.5 |
| Croatia | 162.0 | 495.2 | 221.0 | Luxembourg | 267.5 [y] | 250.5 | 234.7 |
| Cyprus | 202.7 | 112.0 | 157.7 | Malta | 187.4 | 74 | 167.8 |
| Czechia | 92.0 | 87.3 | 174.4 | Netherlands | 535.3 | 267.7 | 265.9 |
| Denmark | 174.1 | 147.7 | 108.1 | Poland | 110.8 | 157.7 | 177.6 |
| Estonia | 36.5 | 100.4 | 87.7 | Portugal | 140.1 | 155.5 | 198.5 |
| Finland | 142.6 | 122.9 | 91.6 | Romania | 38.6 | 45.6 | 59.2 |
| France | 264.1 | 152.4 | 172.7 | Slovakia | 77.8 | 75.1 | 129.3 |
| Germany | 247.4 | 159.6 | 166.5 | Slovenia | 445.3 | 279.8 | 261.8 |
| Greece | 169.5 | 119.1 | 133.3 | Spain | 173.0 | 106.5 | 157.7 |
| Hungary | 76.9 | 96.7 | 150.7 | Sweden | 105.7 | 99.0 | 100.4 |
| Ireland | 656.2 | 857.2 | 1544.9 | | | | |

[z] Fertilizers considered in this table are all products based on nitrogen, potassium, and phosphorus (NPK) elements. This data does not include either animal or vegetative manure (data source: The World Bank—World Development Indicators, [34]). [y] Referring to year 2002, the first available data for Luxemburg.

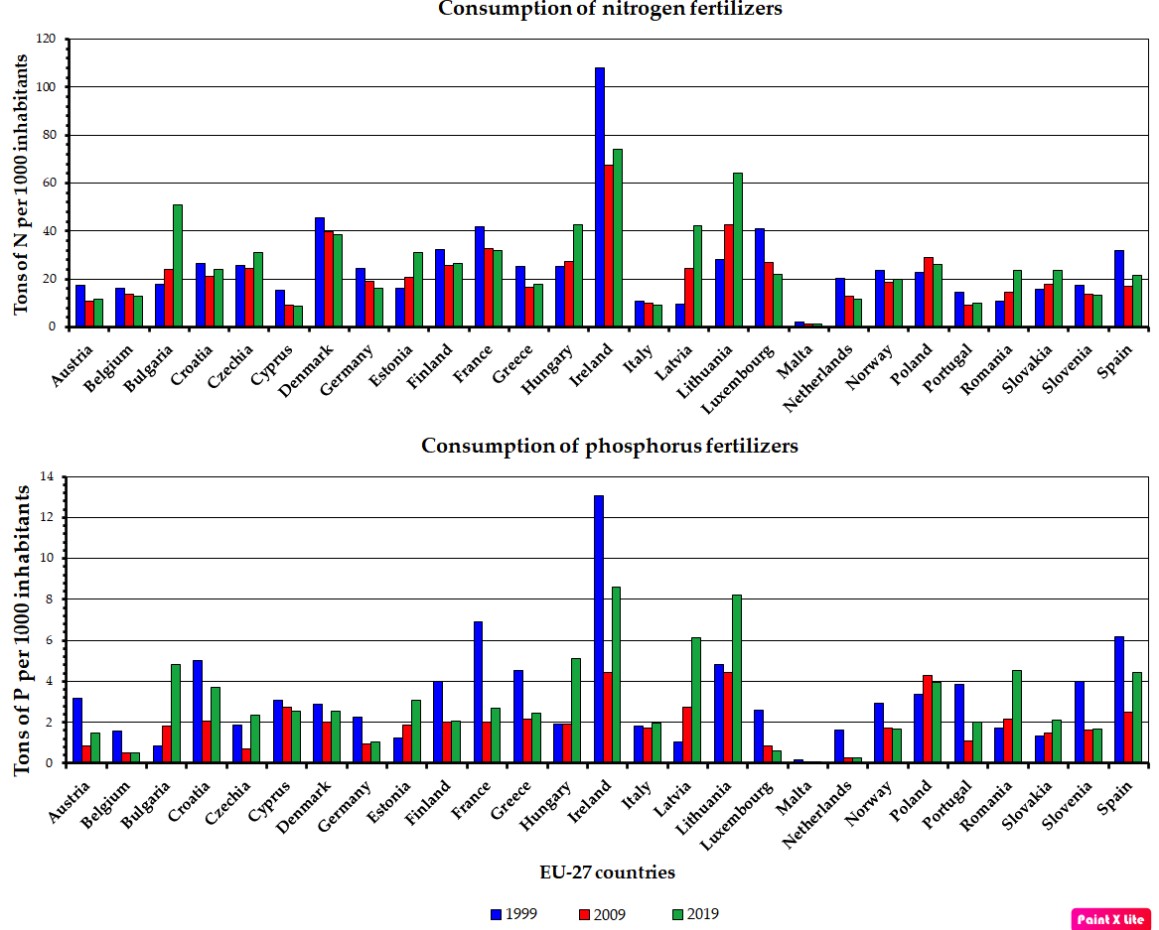

**Figure 1.** Consumption of nitrogen and phosphorus fertilizer in the EU-27 members states in 1999 (blue bars), in 2009 (red bars), and in 2019 (green bars). Tons of N-based fertilizers expressed as tons of N consumed by 1000 inhabitants in the upper graph; tons of P-based fertilizers expressed as tons of P consumed by 1000 inhabitants in the lower graph. Data source: EUROSTAT (Datasets: "Consumption of inorganic fertilizers" [AEI_FM_USEFERT] and "Population on 1 January by age and sex" [DEMO_PJAN]).

The trends of N and P consumption by 1000 inhabitants with respect to more than twenty years ago in the European countries are represented in Figure 1. Here, each country showed a peculiar trend. In most cases, however, and when focusing particularly on 1999 and 2019, the trends were declining (except for several Eastern countries such as Bulgaria, Czechia, Estonia, Hungary, Latvia, Lithuania, Poland, Romania, and Slovakia). This evidence fits well with the increase in the agricultural demand and related land used occurring in these countries. Interestingly, similar trends can be observed for N and P fertilizers.

The limited or absent use of mineral fertilization represents the main feature of organic farming, and several organic farming systems are more energy efficient than their conventional counterparts (although there are some notable exceptions) [35]. Consequently, there is still much to be done, and it is particularly recommended to improve fertilizer (N) management in organic production to ameliorate its energetic and economic performance [36,37]. In 2019, Europe invested 8.5% of the utilized agricultural area (UAA) in organic farming, and is foreseen to increase in coming years. In this general scenario, Italy performed very well with 15.2% of the UAA in organic, and was the fourth country for the highest shares of organic land after Austria (25.3%), Estonia (22.3%), and Sweden (20.4%). Several major EU countries showed lower percentages of UAA under organic cultivation, including Ireland (1.6%), Bulgaria (2.3%), the Netherlands (3.7%), Norway (4.6%), and France and Germany (both with 7.7%) [16,38].

In Italy, the increase in organic agriculture is the major cause of the reduction of total fertilizer distribution, as can be noted in Figure 2, which shows the trend of fertilizer application in Italy from 2003 to 2019, referring to the amounts of fertilizers used in traditional and in organic agriculture, respectively.

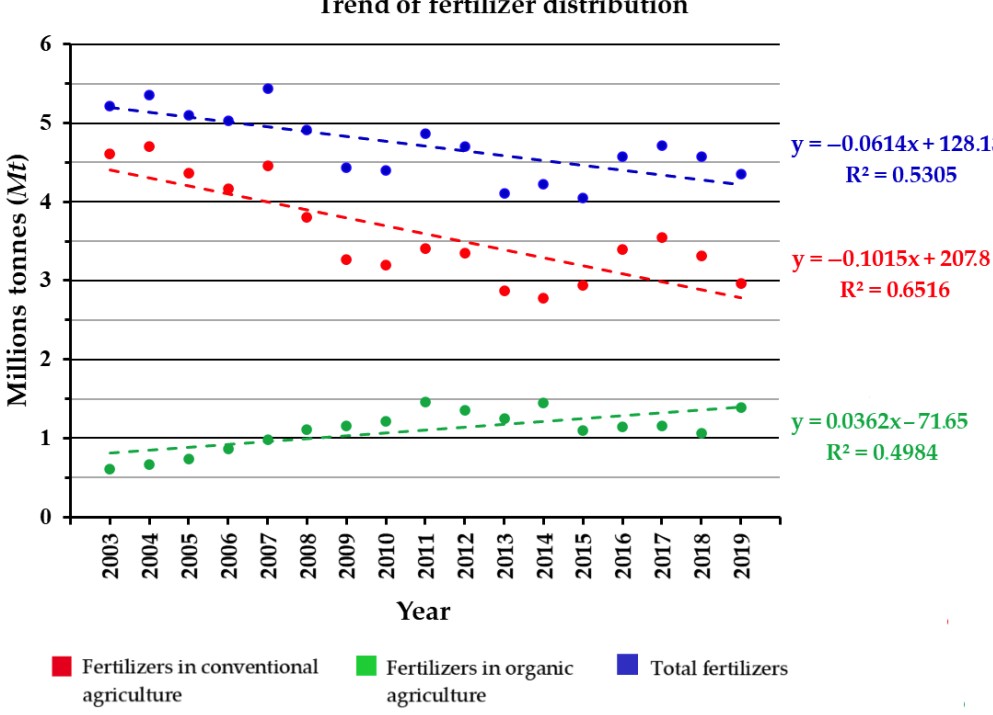

**Figure 2.** Trends of the fertilizer distribution, in million tons of the main nutrient elements, in Italy from 2003 to 2019. The upper blue line is for total fertilizer consumption and is equivalent to the sum of all kinds of fertilizer products used in both conventional (red line) and organic (green line) agriculture. The equation and the R-square value of the linear regression trendlines are displayed on the right of the graph. Total fertilizers include all types of fertilizers reported in Table 3. Data source: ISTAT (https://www.istat.it/en/legal-notice (accessed on 20 June 2020)), [18].

**Table 3.** Use of the different types of fertilizers in Italy, in tons (*t*) of the main nutrient elements in 2009 and 2019.

| | | | 2009 | | | 2019 | | |
|---|---|---|---|---|---|---|---|---|
| | | | Use in Conventional Agriculture | Use in Organic Agriculture | Total | Use in Conventional Agriculture | Use in Organic Agriculture | Total |
| Fertilizers | Mineral fertilizers | Simple — Nitrogen | 1,055,523 | 0 | 1,055,523 | 1,001,488 | 0 | 1,001,488 |
| | | Simple — Phosphate | 122,608 | 564 | 123,172 | 77,458 | 4184 | 81,642 |
| | | Simple — Potassic | 53,693 | 10,792 | 64,485 | 51,701 | 13,344 | 65,035 |
| | | Compound — Binary | 386,801 | 2861 | 389,662 | 270,474 | 2936 | 273,410 |
| | | Compound — Ternary | 452,369 | 0 | 452,369 | 266,974 | 9265 | 276,239 |
| | | Containing meso-elements | 2082 | 3612 | 5693 | 592 | 4345 | 4937 |
| | | Micronutrient fertilizers | 2800 | 10,625 | 13,425 | 3581 | 9399 | 12,980 |
| | Organic fertilizers | | 14,172 | 269,992 | 284,164 | 23,823 | 345,758 | 369,581 |
| | Organo-mineral fertilizers | | 215,660 | 36,060 | 251,756 | 220,221 | 110,957 | 331,178 |
| | TOTAL FERTILIZERS | | 2,305,769 | 334,506 | 2,640,250 | 1,916,312 | 500,188 | 2,416,490 |
| Other products | Amendments | | 835,378 | 763,052 | 1,598,430 | 503,289 | 817,281 | 1,320,570 |
| | Correctives | | 122,723 | 65,683 | 188,405 | 352,509 | 58,254 | 410,763 |
| | Growing substrates | | 9607 | 0 | 9607 | 128,352 | 4663 | 133,015 |
| | Specific action products | | 1348 | 0 | 1675 | 54,947 | 9618 | 64,565 |
| | TOTAL OTHER PRODUCTS | | 969,056 | 828,735 | 1,798,117 | 1,039,097 | 889,816 | 1,928,913 |
| TOTAL FERTILIZERS & OTHER PRODUCTS | | | 3,274,800 | 1,163,240 | 4,438,040 | 2,955,409 | 1,389,994 | 4,345,403 |

(Data source: Italian National Institute of Statistics, ISTAT).

According to the Italian National Institute of Statistic (ISTAT), in 2019 more than 2.4 million tons of fertilizers were applied for agricultural use (or more than 4.5 million tons when including other products, such as amendments, correctives, growing substrates, and specific action products) (Table 3). The amount of mineral fertilizers was about 1.7 million tons. Of these, more than 65% were simple minerals (N, K, P) and more than 30% were more complex minerals. Organic farming use of mineral fertilizers mainly concerned those containing meso and micro-nutrients. In fact, these types of fertilizers applied more in organic farming than in traditional farming. It is worth noting that the total distribution of organic and organo-mineral fertilizers on Italian land was about 0.37 and 0.33 million tons, respectively, meaning that the 93% of total organic fertilizers and the 33% of total organo-mineral fertilizers were applied in organic farming (Table 3). From 2009 to 2019, the use of organic fertilizers in organic farming showed an increase of about 28%, while the use of amendments was just 7%.

Besides a wider adoption of organic farming strategies, the other main reasons leading to the reduction of fertilizer consumption in Italy are related to the rationalization of the use of chemical products in agriculture, as a consequence of the reception of the Council Directive 91/676/EEC (Nitrates Directive) by the Italian Legislative Decree 152/2006 (Environmental Protection Code). The diffusion of the model of bio-agriculture, which enhances the use of amendments (soil improvers), correctives, and sustainable renewable products at the expense of the conventional model of "chemical agriculture", and the reduction of the UAA due to desertification and soil contamination, also contributed to this effect. The Italian National Research Council (CNR) estimates very high percentages of land undergoing or at risk of degradation, particularly in internal rural areas of Southern Italy (up to 70% in Sicily) due to heavy soil erosion. However, desertification represents a major threat that extensively affects the Mediterranean, Central, and Eastern European countries [39].

The significant decrease in fertilizer consumption per hectare of cultivated land, in terms of UAA, that Italy has experienced over recent decades (see Table 2) is strongly related to the steady increase of the land area and the number of producers dedicated to the cultivation of organic goods (Table 4).

**Table 4.** Fertilizer distribution, Utilized Agricultural Area (UAA) and number of farms in conventional and in organic agriculture, in 2010, 2013, 2016, and 2017 in Italy. Total fertilizers include all types of fertilizers (see Table 4).

|  | Unit of Measure | 2010 | 2013 | 2016 | 2017 |
|---|---|---|---|---|---|
| Fertilizers in conventional agriculture | Million tons of the main nutrient/s (Mt) | 3.20 | 2.90 | 3.40 | 3.60 |
| Fertilizers in organic agriculture | | 1.21 | 1.25 | 1.15 | 1.16 |
| Total fertilizers | | 4.40 | 4.11 | 4.58 | 4.71 |
| UAA in conventional agriculture | Hectares (ha) | 12,856,048 | 12,425,996 | 12,598,161 | 12,777,044 |
| UAA in organic agriculture | | N.A. | 961,594 | 1,555,522 | N.A. |
| Total (conventional + organic) UAA | | N.A. | 13,387,590 | 14,153,683 | N.A. |
| Number of farms in conventional agriculture | - | N.A. | 1,471,185 | 1,145,705 | N.A. |
| Number of farms in organic agriculture | | N.A. | 47,075 | 132,299 | N.A. |

(Data source: ISTAT, [18]).

Previously, European Regulation No. 2003/2003 (https://eur-lex.europa.eu/eli/reg/2003/2003/ (accessed on 20 June 2021)) defined the various mineral fertilizers, such as those that provide main nutrients, secondary nutrients, microelements, inhibitors, and calcination substances. In a different way, in Italy, national legislation (Legislative Decree 75/2010) regulates all categories of fertilizers, i.e., mineral fertilizers, organic fertilizers, organic-mineral fertilizers, soil amendments, corrective substances, substrates, and specific action products including biostimulants. The latter are generally extracts of algae, plant or animal hydrolysates, or mycorrhizae, which add substances to another fertilizer, soil

or directly to the plant that assists the absorption of nutrients. Europe is, globally, the largest market for biostimulant products, which is dominated by Germany, followed by Spain and Italy, for innovative products as well as for constant investments in R&D (Assofertilizzanti, Federchimica). On 5 June 2019, a new EU Regulation 2019/1009 of the European Parliament [40] was published to replace No. 2003/2003, specifying all categories of fertilizers at the EU level and introducing new limit values for contaminants, such as cadmium, permitted in each fertilizing product. Finally, this included fertilizers proceeding from recycled or organic materials (in line with a circular economic vision) and biostimulants (as products which enhance plant nutrition processes, independent of the product nutrient content), with the aim of improving properties such as nutrient use efficiency, tolerance to abiotic stress, quality traits, etc., embracing products based on microorganisms.

Regarding pesticides, these are substances that interfere, hinder, or destroy living organisms (microorganisms, animals, and plants), used in intensive industrial-agriculture, and include fungicides, insecticides, herbicides, etc. Pesticide use encompasses 2–4% of the total energy used in crop production [41], and on average the manufacture of pesticides which is a highly complex process takes four to five times more energy per kg than N fertilizer production [42]. Even though nowadays several alternative intervention and prevention techniques are available, pesticide employment in agriculture is still widespread, and laboratory analyses of fresh or processed fruit and vegetable samples often highlight quite high traces (residues) deriving from their use [43]. Good Agricultural Practices (GAPs) should respect the maximum residue level (MRL, in mg/kg) for several pesticides in food commodities and animal feeds as established by the EU Commission.

The use of pesticides and, in general, of chemical products in agricultural soils also has a negative effect on soil biodiversity by altering faunistic and floristic ecosystems [44]. Last but not least, pesticides disrupt soil biotic communities [45,46].

In this complex framework, it is necessary to mention the practice of precision agriculture, with its wide range of new emerging technologies, that allow for precise fertilization based on in-field crop phenotypic performance [47], as well as on soil properties [48], which can be monitored in real-time, leading to notable savings in fertilizer application and an ensuing reduction of the environmental impact of NPK chemicals [49].

As a perspective for agricultural fertilization at the global level, the demand for fertilizer use of nitrogen (N), phosphorus (P), and potassium (K) will reach more than 166 million tons in 2022, with an increase of about 9% with respect to 2016 (FAO, 2019; [50]). In the European Union, changes in annual fertilizer consumption, forecast by Fertilizers Europe for the decade 2016–2026, are projected as -5.0% in nitrogen, +0.7% in phosphate, and +1.8% in potash. In Italy, in recent years, the trend of fertilizer consumption is showing a decrease in line with the increase in organic farming.

## 5. Organic Fertilization

The impoverishment and degradation of soil and the loss of organic substances and microbial diversity constitute a major environmental emergency. Organic substances are a very dynamic component of the soil represented by the plant and animal residues present. Natural organic amendments are used above all, but not exclusively, in organic farming in order to improve the physico–chemical characteristics of soils and increase their fertility. Sometimes, soil amendments present a low nutrient amount but confer a high organic content. However, since organic matter can act as a nutrient reservoir for plants, they often allow for the limiting of the use of fertilizers. With the aim of preserving the soil's organic substances, conservative agricultural practices which rely on reduced soil processing are spreading more and more, and, at the same time, the use of natural organic soil improvers is becoming more common. Some of these exhibit multiple functional characteristics besides fertilization, including the control of plant pathogens. This makes them even more interesting from an energy savings point of view, as they represent a sustainable alternative to synthetic products (with a reduction of indirect agricultural costs). They

require a lower number of annual soil tillages, resulting in fuel-saving for machinery (direct agricultural costs), minor disturbances (mechanical stress) to the soil, and less degradation of the organic substance and subsequent mineralization.

Given that mineral fertilizers represent an expensive component in agriculture and, in parallel, environmental safety is a main focus for human health, any cost-effective natural and renewable products that can replace/reduce the volume of fertilizer needed should be regarded as valuable. As argued in the next sections of this review, compost from biowaste (food waste and green waste), biochar from green biomasses, and neem cake from the waste of neem oil industrial extraction can work very well as soil fertilizers/amendments, enhancing the implementation of a circular economy. With respect to pesticides, compost and even biochar can help retain them in agricultural soils, reducing in this way the pollution of surrounding areas and groundwater [51]. In a different way, due to its prominent constituent azadirachtin, neem cake has emerged as a highly potent bio-pesticide [52,53].

### 5.1. Composting Recycles Organic Waste and Produces a Natural Fertilizer

Biological processes such as composting and anaerobic digestion can add value to the organic fraction deriving from the separate collection of urban waste, which includes mainly residues and food waste. Both composting and anaerobic digestion are based on the degradation of organic matter and occur, respectively, under aerobic and anaerobic conditions. Compost is the end product of the composting process, while biogas and a non-stabilized digestate are the end products of the anaerobic digestion process. Even though the European Commission (2010) agrees that an improved biowaste management "holds an untapped potential for significant environmental and economic benefits" [54], the EU generates about 120–140 million tons of biowaste per year, and only 30–35 of it is recycled into high quality compost and digestate [55] with large differences among the member states, that, in part, reflect their decisions to adopt more or less stringently the EU waste policy and legislation (Waste Framework Directive, WFD, Directive 2008/98/EC on waste). While some countries (such as Austria, Switzerland, Germany, the Netherlands, Belgium, Sweden, and Norway) have been much more involved in separate organic waste collection and in its recycling in recent years, and other countries (such as Italy, Finland, Ireland, Slovenia, Estonia and France) have exhibited an improved attitude toward this sector, there are still several countries that do not show reliable advances in this sector (data obtained from the European Composting Network, ECN). It is worth noting that the very recent "Circular Economy Action Plan", as a part of the European Green Deal, includes revised legislative proposals on waste.

Focusing on Italy, municipal organic waste collection has increased in recent years and, in fact, the percentage of families that performed an accurate wet waste segregation was 36.6% in 1998, 69.8% in 2012, and 83.9% in 2018, with the northern regions playing the major role (data source: ISTAT). According to the Italian Compost Consortium (CIC), in 2017 the annual collection per capita was 108 kg, reaching a total of 6.6 million tons of organic waste. In 2018, the number of plants used for organic waste treatment was 339 including 281 composting plants, 35 plants for integrated aerobic/anaerobic treatment, and 23 anaerobic digestion plants [56]. In the last ten years, this sector has been reinforced, with a 3–4% increase of composting plants per year (CIC), thus providing evidence that this represents an important challenge for the national economy.

In Italy, compost is a fertilizer by law (Legislative Decree 75/2010—Annex 2 and its subsequent amendments and integrations), and it can be produced from green-waste only (green compost), food and green-waste (biowaste compost), or different feedstocks including sludge (sludge compost). The law also establishes the requirements for obtaining good compost and the maximum limits allowed for polluting substances and microorganisms.

The use of compost helps to reduce the phenomena of soil depletion, improving its chemical–physical and biological properties [57], replenishing organic matter with macro and micro-nutrients and making them available for cultivation [58,59], even though some short-term studies have not indicated these effects when low rates of compost are

applied [60]. In addition, compost can improve $CO_2$ sequestration in the soil and reduce nitrogen oxide ($N_2O$) emissions resulting from the administration of chemical nitrogen into the soil [61].

As an organic soil conditioner, it is used in agriculture, horticulture, urban agriculture, and in all of the sectors of agricultural production that involve treatments with eco-compatible substances and do not have negative effects for human health [62,63]. Compost can therefore replace, or in any case reduce, the use of mineral fertilizers with positive consequences on the reduction of energy consumption necessary for their production [64]. Compost has good application prospects in the nursery sector, where the need to reduce the use of peat (a non-renewable, expensive, and largely imported resource) is pressing [65,66]. As an example, the addition of compost obtained from food residues from cafeterias to soilless substrates increased the productivity of lettuce cultures carried out on roofs. This result was further improved by adding biochar to compost [67].

### 5.2. Biochar Can Be Produced by Different Kinds of Vegetal Biomass

Biochar (charcoal produced by pyrolysis of biomass) is another soil improver that can be used alone, but which is even better when used together with a compost (the latter for the sake of adding an organic substance). Biochar is a "secondary" product of the gasification (or pyrolysis) of vegetable biomasses for the production of biofuels, such as syngas and bio-oils, and has a considerable added value given the number and diversity of its possible applications [68]. Biochar can potentially be produced from any biomass. Therefore, its characteristics and the effects that it will play once added to the soil will be strongly dependent on the starting biomass (feedstock), as well as on the reaction conditions (particularly with respect to temperature and pyrolysis time) [69]. Although, theoretically, the use of biochar in conjunction with a compost may increase its beneficial effects [67,70], research is still developing general rules that fit indiscriminately into all environmental situations, and a greater collaboration among researchers and stakeholders should be utilized to move forward the implementation of this technology at a global scale [71].

The variability of the soil, its content with respect to macro and micro-nutrients, the organic substances present, the cultivated agricultural species and variety, the climate, and other secondary factors are all integrated together in a very complex way. Therefore, further studies and experiments on biochar in the field are still necessary for the purpose of optimizing final agricultural performance. A lot of research has been performed and is still underway to define the effects of biochar on plant growth, and it is now clear that the original feedstock plays a key role in assessing effects on plant rhizosphere diversity (which is tightly linked to plant performance). For example, in a comparative study, durum wheat growth exhibited a better performance in soil amended with wheat straw biochar than woody biochar. However, this result was strongly dependent on the environmental conditions including soil. The crop variety (genotype) was also ascertained to play an important role. Thus, to attain the best advantages from the use of biochar, the integrated selection of the more adaptable type of biochar and the cultivar of the crop to be cultivated in a specific agricultural soil can lead to a superior yield [72].

As a BSA, biochar has another advantage. It can bind to pesticides in soil, thus decreasing their leaching. Its further use as a means for soil and water remediation is very appealing and research is in progress concerning this aspect [73]. As for any soil amendment, biochar application rates should be recommended based on previous field trials related to specific soil types and crops. For the moment, a lot of studies have shown a positive influence on crop yield when biochar was used as a one-time application at a rate from 5 to 50 t/ha, together with the required amounts of nitrogen and other nutrients [74]. Looking at the economic feasibility of a wide-scale biochar application in agricultural soil, which is still limited by its rather high cost due in large part to the relevant thermal demand of the pyrolysis process, the production of low-cost biochar on a large scale is still

a challenge [75,76] and lower rates of biochar application (5–10 t/ha) to attain the same agricultural benefits should be a major focus.

In Europe, the leading countries doing biochar research, and consequently on biochar use, are the northernmost regions [77]. As reported in the very recent EU Commission Implementing Regulation 2019/2164 of 17 December 2019 [78], biochar (only from plant material) has been included in the list of authorized fertilizers, soil conditioners, and nutrients for organic production. In Italy, in compliance with the Decree of 22 June 2015 (Update of Annexes 2, 6, and 7 of Legislative Decree No. 75 of 29 April 2010 "Reorganization and revision of the rules concerning fertilizers") biochar became part of the list of fertilizers used in agriculture, thanks to the proposal supported by the Italian Biochar Association (ICHAR). However, given that each biochar presents unique features, it was not appropriate to include all the biochars in the list of usable soil improvers and in fact, according to Italian law, each type of biochar should be characterized and certified before being used. In this regard, to date several biochar certification programs are underway, all on a voluntary basis, open and in progress, with the more accredited ones being the one proposed by the International Biochar Initiative (IBI) and the European Biochar Certificate [79].

For all the above-mentioned reasons, data on biochar consumption at the EU country level are not yet available, nor are data on energy consumption associated with the production chain and use of this product, though it is proving to be very promising to strengthen the circular economy once its production costs are more than halved. Indeed, as also revealed by a recent analysis of the International Biochar Initiative, the market price of biochar in an international scenario varies from 90 to 5000 US $/t, and this very wide gap depends on several reasons including the different degree of development of the economies in different countries.

*5.3. Neem Cake Is Nutrient Rich and Can Replenish Soil Organic Matter*

The neem tree (*Azadirachta indica* A. Juss) is considered to be one of the most promising trees of the 21st century for the development of sustainable food production (WHO/UNEP, 1989). Neem cake is the waste product of neem oil extraction buy industry, and has a high nitrogen content of 2–5% (Table 5), so its use as good low-cost fertilizer is challenging.

**Table 5.** Main nutrient element content as percentage (%) or as part per million (*ppm*) in neem cake.

| | Nutrient Element | Content |
|---|---|---|
| N | Nitrogen | 2–5% |
| P | Phosphorus | 0.5–1% |
| K | Potassium | 1–2% |
| Ca | Calcium | 0.5–3% |
| Zn | Zinc | 15–60% |
| Cu | Copper | 4–20% |
| S | Sulphur | 0.2–3% |
| Mg | Magnesium | 0.3–1% |
| Fe | Iron | 500–1200 ppm |
| Mn | Manganese | 20–60 ppm |

(Data source, the neem encyclopaedia [80]).

The activity and lack of environmental toxicity of neem-based products have been certified by the US Environmental Protection Agency (EPA, 2012, PC code 025006). In accordance with the European Parliament 2019/1009 Regulation [40], the use of neem cake is promising for the improved cultivation of high-quality fruit and vegetable products, in compliance with environmental sustainability and the natural balance of the soil with biostimulant activity. On this point, neem cake's role as biostimulant is related to its role in sustainable nematode management, either by limiting the nematode impact on plant growth or by enhancing host–plant resistance as a consequence of its providing a nutrient source for proper plant growth [81,82]. The multipurpose fertilizer and soil conditioner-nematicidal–insecticidal potential of neem cake could lead to a significant reduction in the

use of mineral fertilizers and pesticides, limiting mechanical soil processing, environmental pollution, and fossil energy exploitation [83,84].

Different types of neem cake products are available on the market depending on the extraction process through which they are obtained. There are neem oil cakes with about 6% oil residue in the cake, and de-oiled neem cakes that present a lower (about 1.5%) oil residue, following an additional extraction process with organic solvent, which can be performed in order to attain a superior oil yield. This last type of neem cake has exhibited a better performance for agricultural use [85]. At the beginning, the trade of neem cake was pushed mainly for its nitrogen content and macro and micro-nutrient composition (Table 5). Now that its effectiveness as a biostimulant and its other polyvalent properties have been assessed, a further extension of neem cake exploitation can be foreseen.

As with compost and biochar, neem cake is not a single well-characterized product, and its characteristics are variable and strongly depend on geographical origin and the seed drying and extraction process. Indeed, the quantitative and qualitative chromatographic profiles of different neem cakes exhibit a variable content of secondary metabolites (as limonoids), out of those which are specific of the *Azadiratha indica*, which persist in the by-product after the extraction process [86]. Hence, further research efforts are mandatory in order to characterize the best quality requirements of neem cake in accordance with its final intended use, in order to stabilize its pesticide, nematicide, and biostimulant functions.

Since neem cake is a natural product without any chemical modification, it is not subjected to the EU REACH (Registration, Evaluation, Authorization, and Restriction of Chemicals) regulation for the protection of human health and environment from the risks that can be posed by chemicals.

When applied as a soil amendment, it binds several macro and micro-nutrients, allowing their controlled release and limiting their loss by leaching [87,88]. Thus, neem cake is also highly recommended in protected agriculture, particularly for growing various vegetables and strawberries, demanding frequent irrigations, where it is used in a 1:4 ratio with nitrogen fertilizer or, according to organic agriculture practices, in the same ratio with algae, organic humates, and seed flours such as the castor cake (another oil seed cake used as a soil conditioner). In protected horticulture, a common neem cake application rate is 0.24–0.4 t/ha, while in the field it is 0.4–0.6 t/ha. In order to take advantage of its insecticidal and nematicidal activity, practical experience has suggested dosages of 1–2 t/ha. In organic farming, neem cake is also very useful in pre-sowing and pre-planting, where its efficacy in the prevention of nutritional imbalances by avoiding the accumulation of nitrates in plant tissues has been proven [89]. Neem cake also has important effects on soil microbiome communities, increasing microbial population, and on soil carbon sequestration [78]. Furthermore, the gradual availability of nitric nitrogen to plants, which is conferred by the neem cake in the soil, promotes optimal iron absorption. With regard to different chronic forms of the deterioration of arboreal fruit trees caused by fungal infections and iron deficiency, neem cake as well as treatment with iron-chelate in open field trials showed a positive influence in mitigating such diseases and at increasing leaf chlorophyll content [90]. In addition, acting as a natural nematicide, neem cake can be used in soil biofumigation [91], replacing methyl bromide (known to be a stratospheric ozone depleting substance) in a sustainable way and without negative environmental impact.

Worthy of note has been the acknowledgment received through the international "Special Award for Organic Best Practices" for farming innovation by the project "Neema-grimed" [92]. As a corollary to the success gained at the EXPO exhibition, the outcomes of this research have enhanced the diffusion in western countries of the use of neem cake as a novel product in organic farming. The export of neem-based products from developing countries to Europe would enhance commercial exchanges between them and the possibilities of cooperative projects, leading to job creation in the agricultural neem value chain. Available and accessible data on the neem market in 2015 at the global level was $653.7 million in revenue. Moreover, for the forecast period of 2016–2020, it was expected to show a compound annual growth rate (CAGR) of 16.3%. However, it showed instead an

increase of over 17%, and is expected to garner a revenue of $ 2.04 billion by 2022 [93,94]. The main factors fostering the growth of the market of neem extract can be identified in the growing awareness about the side effects related to the extensive use of chemicals, and in the willingness to replace them with natural products, such as the neem cake, together with the favorable regulatory environment. In Europe, neem extract application as a biopesticide is showing a larger share than as fertilizer and, in this respect, the withdrawal from the market of several toxic chemicals (as for example temephos and 1,3-dicloropropene) will sustain the use of neem cake as a multipurpose biofertilizer not only in organic but also in traditional pest management.

## 6. Conclusions

The reduction in the short term of the high energy demand and emission of pollutants proceeding from fertilizer manufacturing is crucial. Energy costs may be substantially decreased by embracing energy efficient solutions, such as the energy efficiency gains in ammonia production over the past 40 years which have led to an average 30% of energy savings per ton of ammonia produced. Moreover, precision agriculture techniques are becoming relevant and very useful because they are an ideal management of fertilizers and water supplying only the required amounts.

On the other side, organic farming is environmentally sustainable, with positive potential implications for economics, but its energy demand is lower than conventional farming and its capability for providing food for the increasing global demand has not yet been assessed. Nonetheless, the farmer's raised awareness of organic fertilizers together with governmental policy and support are guiding the growth of the organic fertilizer market, and a prognosis for Europe forecasts a compound annual growth rate (CAGR) of 4.2% from 2017 to 2023 [95]. The European Commission strongly supports circular economy schemes that enhance the exploitation of natural renewable resources, such as biowastes, which have a high potential to provide the alleviation of major environmental concerns. In order to react to the increasing food demand, research in this area must be recognized as a high priority and needs the development of dedicated international networks in order to share all details on field trial conditions and results, since the increasing application of natural organic amendments into agricultural soils will be beneficial for soil, the environment, food safety, and will result in lower amounts of mineral fertilizer application.

**Author Contributions:** Conceptualization, A.L. and S.M.; formal analysis, A.L. and G.G.; resources, C.A.C.; data curation, A.L. and C.B.; writing—original draft preparation, A.L.; writing—review and editing, A.L., G.G., C.A.C., C.B. and S.M.; supervision, A.L.; funding acquisition, C.A.C. All authors have read and agreed to the published version of the manuscript.

**Funding:** Supported by the Italian National Funding "Electric System Research Programme 2019–2021" from the Ministry of Economic Development.

**Institutional Review Board Statement:** Not applicable.

**Informed Consent Statement:** Not applicable.

**Data Availability Statement:** Not applicable.

**Acknowledgments:** We thank to M. Marani and G. Puglisi (ENEA) for their valuable and helpful suggestions.

**Conflicts of Interest:** The authors declare no conflict of interest.

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
