# Peer review of "A Narrative Review of the Facts and Perspectives on Agricultural Fertilization in Europe, with a Focus on Italy"

_horticulturae, doi:10.3390/horticulturae7060158_

Round 1
Reviewer 1 Report
The paper “A narrative review on the facts and perspectives of the agricultural fertilization in Europe, with a focus on Italy" shows an abstract that gives a clear idea of the content of this work.
The introduction is relatively tight is length. The methods are descriptive, as befits a review. The Energy requirement for the industrial production of fertilizers section is quite complete, as are the Consumption of fertilizers in traditional and in organic agriculture and the Organic fertilization. Conclusions make special reference to the different benefits of increasing application of natural organic amendments into agricultural soils. Finally, maybe the references are a bit long, but I think that is what corresponds to a review article.
Reviewer 2 Report
Dear Authors
The work is interesting and presents the current problems of agriculture and horticulture. However, the authors did not avoid flaws and errors:
1) The introduction should refer to the aim of the work, as well as contain hypotheses / research hypotheses that had to be verified in the work, this element is missing.
2) The section "Material and methods" is poorly described and does not inform the reader how they want to prove the superiority of organic fertilization over mineral fertilization, what methods of statistical calculations were used, how were, for example, fertilization trends calculated?
2a) Why the authors do not provide new sources of energy consumption in the production of herbicides, fungicides and organic fertilizer additives (effective microorganisms)
Ad 4. ​​"Fertilizer consumption". Fig. 1. Lack of determined trends, moreover, the period of 10 years is too short to determine the rate of decrease or increase in the level of mineral fertilizer consumption.
I do not agree with the statement that organic farming is less energy-intensive - there is no coverage in economic calculations, in statistical data. Organic farming, due to the lack of the use of chemical plant protection products, requires much greater mechanical protection against weeds, and the use of "ecological fertilizer additives" requires multiple spraying (8-10 times), therefore the energy costs are higher, there is a lack of fuel and water increasingly due to global warming.
The authors did not bother to calculate the costs of using corrective measures in organic farming (e.g. effective microorganisms) incurred for their production and consumption (8-10 treatments should be performed to notice the effect of these treatments), i.e. related mainly to the use of large amounts of sprayed water ( 300-400 liters of water per 1 ha for each treatment). Or by citing such data from the literature.
Fig. 2. On the basis of what calculations did the authors determine the fertilizer consumption trend?
The trend should be calculated on the basis of the regression analysis and the coefficient of determination should be reported.
5) "Mineral fertilizers based on macro- and micronutrients in organic farming are more widespread than traditional" - refer to this sentence: Is there a misinterpretation here?
6) Neem cake from industrial neem oil extraction waste is imported from India. Will European agriculture in the future be based on imported organic fertilizers?
7) The authors devote a lot of space to biochar, but do not provide any data on this subject, apart from a limited literature
8) The conclusions are not very specific, they are a kind of further part of the discussion and not a summary. Conclusions should be summarizing and generalising. It should be shortened and clarified.
9. Not enough literature on alternative fertilizer forms
Reviewer 3 Report
The article summarizes a large amount of information on the use of fertilizers in Europe in general and in Italy in particular.The authors correctly raise the question of the possible negative consequences arising both in the production and in the use of mineral fertilizers. They rightly point to the high energy consumption of production, especially of mineral nitrogen fertilizers. In this regard, the urgency of reducing the use of mineral fertilizers and the production of organic agricultural products, using organic and non-traditional fertilizers as much as possible, is emphasized. As a wish, it should be pointed out that the authors did not focus on the disadvantages of organic fertilizers, biochar and, in general, the possibility of a significant decrease in labor productivity in agriculture with the widespread transition to organic farming (if we adhere to the term "organic farming" in our country, Russia). . An interesting fact is that in the "organic agriculture" of Italy, a huge amount of nutrients are used in the composition of mineral and organic fertilizers (about 20.7% of the total), although the share of land occupied by organic agriculture in 2019 was from the total area of ​​agricultural land is only 15.2%. In addition, it is noted that mineral fertilizers based on meso- and microelements are more widespread in organic agriculture than in traditional agriculture: 13744 and 4173 tons, respectively (Table 4). In general, organic agriculture in Italy in 2019 used (including taking into account non-traditional sources of nutrients - “other products”) about 32% of the total nutrients used in the country's agriculture! All this indicates that the representations of worthy authors are from the representations of our country on this issue.
